# Extracts of *Phyllostachys pubescens* Leaves Represses Human Steroid 5-Alpha Reductase Type 2 Promoter Activity in BHP-1 Cells and Ameliorates Testosterone-Induced Benign Prostatic Hyperplasia in Rat Model

**DOI:** 10.3390/nu13030884

**Published:** 2021-03-09

**Authors:** Kwang Hoon Song, Chang-Seob Seo, Won-Kyung Yang, Hyun-O Gu, Ki-Joong Kim, Seung-Hyung Kim

**Affiliations:** 1Herbal Medicine Research Division, Korea Institute of Oriental Medicine, Yuseong-daero 1672, Yuseong-gu, Daejeon 34054, Korea; csseo0914@kiom.re.kr; 2Institute of Traditional Medicine and Bioscience, Daejeon University, Daejeon 34520, Korea; ywks1220@dju.kr (W.-K.Y.); rich1169@naver.com (H.-O.G.); cowking1122@naver.com (K.-J.K.); sksh518@dju.kr (S.-H.K.)

**Keywords:** *Phyllostachys pubescens*, benign prostatic hyperplasia, 5α-reductase type 2, dihydrotestosterone

## Abstract

Benign prostatic hyperplasia (BPH) is the most common symptomatic abnormality of the human prostate characterized by uncontrolled proliferation of the prostate gland. In this study, we investigated the effect of bamboo, *Phyllostachys pubescens,* leaves extract (PPE) on human 5α-reductase type 2 (SRD5A2) gene promoter activity in human prostate cell lines and the protective effect of PPE on a testosterone-induced BPH rat model. PPE repressed human SRD5A2 promoter activity and its mRNA expression. The rats treated with PPE for 4 weeks showed a significantly attenuated prostate weight compared to vehicle control. PPE-treated rats also showed reduced serum dihydrotestosterone, testosterone, prostate-specific antigen, and SRD5A2 levels by testosterone injection. Quantitative real-time polymerase chain reaction showed that PPE treatment significantly decreased mRNA expression of SRD5A2, androgen receptor (AR), proliferating cell nuclear antigen (PCNA), and fibroblast growth factor 2 compared with the vehicle-treated, testosterone-injected rats in the prostate. Furthermore, PPE treatment showed reduced AR, PCNA, and tumor necrosis factor alpha expression in the prostate via immunohistofluorescence staining. In conclusion, oral administration of PPE prevented and inhibited the development and progression of enlarged prostate lesions in testosterone-induced animal models through various anti-proliferative and anti-inflammatory pharmacological effects and induced suppression of SRD5A2 gene expression.

## 1. Introduction

Benign prostatic hyperplasia (BPH) is the most common symptomatic abnormality of the human prostate that affects older men [1]. Many men over the age of 40 have histologically identifiable BPH, and the prevalence increases with age, leading to impaired urination [2]. The main treatment for an enlarged prostate is prostatectomy through laparotomy or urethral resection, but it is often associated with a number of complications, including urinary tract infections, sexual dysfunction, or bleeding [3,4].

While studies to develop a clear and detailed pathogenesis of BPH are ongoing, previous studies have shown that male hormones such as androgens, testosterone, and specifically dihydrotestosterone (DHT) are linked to the onset of BPH [5]. The typical role of 5α-reductase is to metabolize testosterone to 5α-dihydrotestosterone, the most potent androgen [6]. 5α-reductase promotes the 5α-position reduction of 4-ene-3-keto steroids such as testosterone, 4-androstenedione, and progesterone [7]. DHT performs important physiological functions such as differentiation of the male external genitalia and secondary growth during puberty, especially in men, but it has been demonstrated to be involved in androgen-dependent diseases [8,9]. There are diseases such as prostate hypertrophy, prostate cancer, and skin diseases such as male hair loss, seborrheic dermatitis, and acne [1,10]. To prevent or treat these diseases, inhibiting DHT generation in target cells is thought to be effective, and various 5α-reductase enzyme inhibitors such as finasteride and dutasteride have been developed and are used in the treatment of BPH [6]. However, these drugs are known to cause adverse effects such as erectile dysfunction, loss of libido, myopathy, and chest pain [11,12].

Recent research suggests that herbs used in oriental medicine can effectively treat BPH or inhibit the occurrence of BPH [13,14,15]. These herbal extracts contain alkaloids, flavonoids, isoflavones, saponins, and other phenolics. With increasing interest in the use of functional foods and dietary supplements for health care, saw palmetto extract is widely used as a functional food to suppress prostate enlargement [16]. However, a product extracted from saw palmetto only had a placebo effect in improving the symptoms of an enlarged prostate and urinary tract and it did not improve the symptoms of an enlarged prostate [16,17]. Therefore, the development of functional food materials with safety and functionality is required.

Bamboo is known to be effective against obesity, and it also relaxes the mind and helps relieve stress, insomnia, and forgetfulness [18,19]. It is also known for its anti-inflammatory, antioxidant, anti-osteoporosis, and anti-wrinkle effects, and it is helpful for hypertension, arteriosclerosis, and activating innate immunity [20,21]. Bamboo leaves are a medicinal herb containing various active ingredients such as flavonoids, phenolic acids, and coumarin lactone [22]. However, the efficacy of bamboo leaves for BPH has not yet been studied. Therefore, in this study, we evaluated the effect of bamboo, *Phyllostachys pubescens,* leaves extract (PPE) on SRD5A2 gene expression in human prostate cell lines and a testosterone-induced BPH rat model.

## 2. Materials and Methods

### 2.1. Luciferase Reporter Constructs

Human SRD5A2 promoter fragments from -1642 to -1 bp were constructed in the pGL3 luciferase vector. A human genomic fragment spanning from -1642 to -1 bp of SRD5A2 was amplified by PCR using the following oligonucleotides: forward 5′- CCCGGGCTCGAGATCTGAAACGGCTATGATGGCT-3′ and reverse 5′- CCGGAATGCCAAGCTTCGCGCCGTGTTCCTCGCC-3′; the fragment was then digested with BglII and HindIII and was infused to the PGL3 basic vector between the BglII and HindIII sites, and the construction of the amplified DNA sequences was verified by DNA sequencing.

### 2.2. Preparation of Phyllostachys pubescens Extract (PPE)

Dried leaves of *P. pubescens* were provided from Nova K-Med Co. Ltd. (Daejeon, Korea). The origin of the sample was confirmed by Dr. Goya Choi, the Herbal Medicine Resources Research Center, Korea Institute of Oriental Medicine (KIOM; Naju, Korea). The voucher specimen (2020PPL) has been deposited at the Herbal Medicine Research Division, KIOM. To prepare PPE, dried *P. pubescens* leaves were extracted with 80% ethanol for 3 h, the extract was filtered, and the organic solvent was removed. Then, it was freeze-dried to obtain a powder sample.

### 2.3. Cell Culture

Human prostate epithelial cells, BPH-1 cells, were obtained from Creative Bioarray (Shirley, NY, USA). The cells were maintained in an RPMI1640 (Life technologies, Carlsbad, CA, USA), supplemented with 1% penicillin/streptomycin (Gibco, Grand Island, NY, USA) and 20% heat-inactivated fetal bovine serum (Gibco by Life Technologies, USA) at 37 °C in an atmosphere containing 5% CO_2_.

### 2.4. Transient Transfection and Luciferase Reporter Assay

For the luciferase reporter assay, BPH-1 cells were plated in 48-well plates 24 h before transfection; then, the cells were transfected with the luciferase reporter using the TranslT-LT1 transfection reagent (Mirus Bio, Madison, WI, USA) according to the manufacturer’s instructions. For reporter quantitation, cell lysates were assayed using the Luciferase Assay System (Promega, Madison, WI, USA) and a TriStar LB941 Luminometer (Berthold technologies, Bad Wildbad, Germany) according to the manufacturer’s instructions. 

### 2.5. RNA Isolation and Quantitative Real-Time PCR (Q-PCR)

Total RNA was isolated using the RNeasy Mini Kit (Qiagen, Hilden, Germany) according to the manufacturer’s instructions. To detect relative mRNA expression, reverse-transcription and Q-PCR were performed with TaqMan probes (Life Technologies, USA) as described previously [23] or with SYBR Premix (Takara, Kusatsu, Japan). Relative amounts of cDNA were calculated by using the relative quantification (ΔΔCt) method. Experiments were repeated at least three times with different cell preparations. The sequences of synthesized PCR primer sets (Bioneer Co. Ltd., Seoul, Korea) for rat 5a-reductase type 2 were 5′-GACCACAGGCGAGATGCAGA-3′ and 5′-TGTGTTTCCCGTAACTGGCG-3′; for rat proliferating cell nuclear antigen (PCNA), they were 5′-CAATTTCTAGCAACGCCTAAGAT-3′ and 5′-AAGAGGAAGCTGTGTCCATAGAG-3′; for rat androgen receptor (AR), they were 5′-CAAAGGGTTGGAAGGTGAGA-3′ and 5′-GAGCGAGCGGAAAGTTGTAG-3′; and for rat fibroblast growth factor (FGF2), they were 5′-GAACCGGTACCTGGCTATGA-3′ and 5′-CCGTTTTGGATCCGAGTTTA-3′. The relative abundance of each transcript was normalized to that of ubiquitin C or GAPDH.

### 2.6. Cytotoxicity Assay

BPH-1 cells were seeded into 48-well plates (2 × 10^4^ cells/well) and incubated for 24 h. The following day, a medium containing PPE (at concentrations ranging from 10–300 μg/mL) was added. Twenty-four hours later, the cell proliferation rate was determined using the CellTiter 96^®^ AQueous One Solution Cell Proliferation Assay (Promega, Wisconsin, WI, USA), according to the manufacturer’s protocol as previously described [23].

### 2.7. Animals, BPH Induction, and PPE Administration

Male Sprague-Dawley rats (6 weeks old) were purchased from Dae Han Bio Link Co., Ltd. (Chungcheongbuk-do, Korea). After a 1-week acclimatization period, the animals were housed and kept under a controlled temperature (22 ± 2 °C) and humidity (55 ± 15%), on a 12-h light/dark cycle. All animal procedures were performed in accordance with the guidelines for the Care and Use of Laboratory Animals developed by Institute of Laboratory Animal Resources of the National Research Council and were approved by the Institutional Animal Care and Use Committee of Daejeon University (DJUARB2020-017) in Daejeon, Korea.

To prevent the effects of endogenous testosterone, rats in all groups except the control (CON) rats group underwent bilateral orchiectomy 2 days before testosterone propionate (Sigma-Aldrich, St Louis, MO, USA) injection. For BPH studies, castrated rats were randomly divided into 5 groups of 8 animals, with equal distribution of body weight. Prostate hyperplasia was induced by subcutaneous injection of testosterone propionate (5 mg/kg) daily for 4 weeks. During the 4 weeks of the prostatic hyperplasia induction period, three of the groups were treated with three oral doses (50, 100, and 200 mg/kg) of PPE, while the positive drug control group received finasteride (100 mg/kg).

Rats were fasted overnight prior to being sacrificed, with blood samples collected via cardiac puncture of anesthetized rats. The serum was separated from blood cells via centrifugation and was stored at −80 °C until analysis. The prostate was immediately excised, weighed, and stored at −80 °C. Other tissues were also excised and rinsed with phosphate buffered saline (PBS) to remove blood from the tissues, flash-frozen in liquid nitrogen, and then stored at −80 °C until analysis.

### 2.8. Histological Analysis

For histopathological examination, the prostate was fixed in 10% formalin and prepared into paraffin blocks. Sections from the blocks with a thickness of 4 μm were stained with hematoxylin and eosin (H&E) and observed with an optical microscope.

### 2.9. Serum Analysis

Serum was separated from the blood via centrifugation at 1500× *g* for 10 min. The content of testosterone, DHT, prostate-specific antigen (PSA), and SRD5A2 in the serum of each group was analyzed by ELISA kit (Mybiosource, San Diego, CA, USA) according to the manufacturer’s protocols. Serum GPT (glutamate pyruvate transaminase, also known as alanine aminotransferase, ALT) and GOT (glutamate oxaloacetate transaminase, also known as aspartate transaminase, AST) were measured using an automatic clinical chemistry analyzer (XL-200, Erba Diagnostics Mannheim, Germany) using 100 μL of heart blood serum.

### 2.10. Immunohistofluorescence Staining

To perform immunohistofluorescence staining, the extracted prostate tissue was cut into a 20-µm thick section at −20 °C using a frozen sectioner and then attached to the slide. To fix the tissue, 4% paraformaldehyde (Sigma-Aldrich, St Louis, MO, USA) and 4% sucrose (Sigma-Aldrich, St Louis, MO, USA) were added and fixed at room temperature for 45 min. 10 mM Glycine (Sigma-Aldrich, St Louis, MO, USA) was added to PBST (a solution containing 0.1% Triton X-100 in PBS) and washed three times for 5 min. Thereafter, 0.5% NP-40 was added to PBS and reacted for 30 min at room temperature. After washing for 5 min with PBST, a blocking buffer with 5% goat serum (Biowest, Riverside, MO, USA), 5% horse serum (Biowest, Riverside, MO, USA), and 3% bovine serum albumin (Sigma-Aldrich, St Louis, MO, USA) was prepared in PBS-T solution. The blocking reaction was carried out overnight at 4 °C with the solution. The primary antibody was diluted in blocking buffer, reacted at room temperature for 4 h, and washed three times with PBST for 10 min each. After, the secondary antibody was also diluted in a blocking buffer, reacted for 2 h at room temperature under conditions in which light was blocked, and washed once for 10 min with PBST. Nuclear staining was incubated with Hoechst 33258. Images were acquired using a fluorescence microscope (Nikon, Japan) and ACT-1 software. Primary antibodies used for staining were AR (SCBT, Dallas, TX, USA, 1:400), PCNA (SCBT, Dallas, TX, USA, 1:400), and TNFα (SCBT, Dallas, TX, USA, 1:400). Secondary antibodies used were fluorescein goat antimouse IgG antibody (H + L) (Invitrogen, Carlsbad, CA, USA; 1:400) and rhodamine goat antirabbit antibody (Invitrogen, Carlsbad, CA, USA; 1:400).

### 2.11. Chemicals and Reagents

Five reference standard compounds were purchased from commercial suppliers: chlorogenic acid (99.6%) from Acros Organics (Pittsburgh, PA, USA), isoorientin (98.5%) from Shanghai Sunny Biotech (Shanghai, China), orientin (99.1%) and isovitexin (99.3%) from Biopurify Phytochemicals (Chengdu, China), and 4-hydroxycinnamic acid (99.2%) from Wako Chemicals (Osaka, Japan). Solvents (methanol, acetonitrile, and water) used were HPLC-grade and purchased from JT Baker (Phillipsburg, NJ, USA). Formic acid and dimethyl sulfoxide used were ACS reagent grade and purchased from Merck KGaA (Darmstadt, Germany).

### 2.12. HPLC Analysis of Five Marker Components in P. pubescens Leaves

The HPLC analysis of two phenylpropanoids (chlorogenic acid and 4-hydroxycinnamic acid) and three flavonoids (isoorientin, orientin, and isovitexin) for quality evaluation of *P. pubescens* leaves was performed in a Shimadzu Prominence LC-20A series (Kyoto, Japan) equipped with two pumps (LC-20AT), an online degasser (DGU-20A3), a column oven (CTO-20A), automatic sample injector (SIL-20A), and a photodiode array detector (SPD-M20A). Five marker analytes were separated by using Gemini C18 (Phenomenex, 4.6 × 250 mm, 5 μm, Torrance, CA, USA) maintained at 40 °C. The mobile phases were eluted by gradient condition with distilled water (solvent A) and acetonitrile (solvent B); both contained 0.1% (*v*/*v*) formic acid. The gradient elution condition of the mobile phase was as follows: 5%–60% B for 0–20 min, 60% B for 25–30 min, and 60%–5% B for 30–35 min. The flow rate and injection volume of standard and sample solutions were 1.0 mL/min and 10 μL, respectively. All data were acquired and processed by LabSolution software (version 5.53, SP3, Kyoto, Japan).

### 2.13. Statistical Analysis

Statistical analysis was performed using GraphPad Prism 7 (GraphPad Software, San Diego, CA, USA). Normality of the distributions was evaluated using Shapiro–Wilk’s test. As the samples followed normal distribution, between-group differences were evaluated using a one-way analysis of variance (ANOVA), followed by Dunnett’s test to identify significant differences between the BPH and CON groups. Significant *p*-values are reported as follows: * and #, *p* < 0.05; ** and ##, *p* < 0.01; and *** and ###, *p* < 0.001.

## 3. Results

### 3.1. PPE Suppresses Human 5α-Reductase Type 2 Promoter Activity in the Human BPH Epithelial Cell Line BPH-1

We constructed a human 5α-reductase type 2 (SRD5A2) gene promoter containing reporter luciferase to screen for substances that can be controlled at the stage of the SRD5A2 gene expression, not to search for substances that inhibit the enzyme activity of 5α-reductase. Since natural extracts have long been recognized as a potential source of treatment, cell-based reporter screening tests have been used to select extracts that inhibit human SRD5A2 promoter activity in BPH-1 cells.

To identify these extracts, BPH-1 cells were plated on the 48-well plate, delivered with the human SRD5A2 promoter reporter, and treated with 200 natural extracts. After 24 h of incubation, luciferase activity was measured. We focused on herbal extracts that inhibited the SRD5A2 promoter activity by less than 50% compared to the control. Initial hits were retested according to dose (Figure 1A). We identified that PPE, an herbal extract, inhibits more than 50% of the human SRD5A2 promoter luciferase activity from this screening.

To further confirm whether PPE represses the human SRD5A2 gene at a transcriptional level, SRD5A2 mRNA levels in BPH-1 cells were measured by quantitative real-time PCR (Q-PCR). Seeding BPH-1 cells in 6-well plates and treatment with PPE (3, 10 and 30 μg/mL) for 24 h resulted in reduced SRD5A2 mRNA expression in a dose-dependent manner (Figure 1B).

The cytotoxicity of PPE was verified using an MTS assay, which showed that treatment with PPE at concentrations up to 300 μg/mL for 24 h did not significantly affect BPH-1 cell viability (Figure 1C).

### 3.2. Effects of PPE on Prostate Weights 

To further investigate whether PPE has a protective effect on testosterone-induced BPH in vivo, PPE was administrated in three different doses (50, 100, and 200 mg/kg/day) for 4 weeks. Consistent with previous reports, rats injected with testosterone for 4 weeks induced prostatomegaly (Figure 2A,B). Prostate weight in testosterone-injected rats was about 2-fold greater than that of the control group rats, but treatment with PPE (100 and 200 mg/kg/day) significantly attenuated prostate weight compared to the testosterone-injected group (BPH) alone. Positive controls, finasteride (1 mg/kg/day), also showed an inhibitory effect on prostate weight compare to the BPH group. We also analyzed the prostatic index which was calculated as the ratio of prostate weight to body weight (Figure 2C). PPE (100 and 200 mg/kg/day) and finasteride (1 mg/kg/day) also significantly inhibited the testosterone-induced prostatic index. 

We further confirmed the inhibitory effects of PPE on prostatomegaly by histological analysis (Figure 2D,E). In the prostate tissue of the control group (CON), one layer of low columnar epithelial cells constitutes the luminal endocrine cells, and the cavity is filled with a light eosinophilic substance, but the prostate epithelial tissue thickness was increased in the BPH group that caused prostate enlargement with testosterone. On the other hand, the PPE-treated group and finasteride group showed reduced epithelial tissue thickness compared to the BPH group. These data indicated that PPE was effective in reducing prostate enlargement in testosterone-induced rats.

### 3.3. Effects of PPE on Serum Prostate Parameters Levels

The finding that PPE supplementation ameliorated prostatomegaly in BPH rats prompted us to evaluate the change in serum prostate parameters (Figure 3). The serum prostate parameters, DHT, testosterone, PSA, and SRD5A2 levels, were significantly increased in BPH group rats compared to control group rats. However, PPE treatment remarkably inhibited these serum prostate parameter levels compared to the BPH group in a dose-dependent manner. These results confirmed that PPE treatment restored the normal form of the prostate in BPH by reducing the increase in serum DHT, testosterone, PSA, and SRD5A2 levels by testosterone injection.

### 3.4. Analysis of SRD5A2, AR, PCNA, and FGF2 mRNA Levels in Prostate

We examined the effects of PPE on SRD5A2, AR, PCNA, and FGF2 mRNA expression in prostate tissue via Q-PCR (Figure 4). Injection of testosterone (BPH) markedly increased the mRNA expression of SRD5A2, AR, PCNA, and FGF2 compared to the CON group. PPE treatment significantly decreased mRNA expression of SRD5A2 (PPE, 100 and 200 mg/kg/day), AR (PPE, 200 mg/kg/day), PCNA (PPE, 100 and 200 mg/kg/day), and FGF2 (PPE, 200 mg/kg/day). Significant change in the mRNA levels of SRD5A2, AR, PCNA, and FGF2 was also observed in the finasteride-treated group. Taken together, PPE was effective to prevent testosterone-induced BPH in rats not only by inhibiting SRD5A2 expression but also by inhibiting AR-dependent prostate proliferation.

### 3.5. Effect of PPE on AR, PCNA, and TNFa Expression in Prostate

In addition, we examined the effects of PPE on androgen-related gene (AR, PCNA, and TNFα) expression in prostate tissue via IHF. As shown in Figure 5, dramatically increased AR, PCNA, and TNFα-positive cells in the epithelium prostate by testosterone injection (BPH) were observed. However, the testosterone-mediated upregulation of AR, PCNA, and TNFα expression were reduced significantly in the PPE- and finasteride-treated groups.

### 3.6. Hepatotoxicity of PPE in Rat BPH Model

To address the effect of PPE on liver damage, serum enzyme activities of GTP and GOT were evaluated (Figure 6). The serum GTP and GOT levels were not significantly changed in BPH-group rats compared to the control-group rats. By confirming the decrease in GPT and GOT in the PPE treatment group, we speculated the possibility that PPE could be helpful in reducing hepatotoxicity.

### 3.7. Quantification of Five Marker Components in P. pubescens Leaves

In the established HPLC assay, the peak of each compound was identified by comparing it with the retention time and UV spectrum of each reference standard compound. All components were separated within 20 min and a representative HPLC chromatogram was shown in Figure 7. Five marker components (chlorogenic acid, isoorientin, orientin, isovitexin, and 4-hydroxycinnamic acid) were simultaneously analyzed in *P. pubescens* leaves samples by established the HPLC-analysis assay. Quantification of these analytes was conducted at 315 nm for 4-hydrocinnamic acid, 325 nm for chlorogenic acid, 335 nm for isovitexin, and 350 nm for isoorientin and orientin. The amounts of the five marker components in the lyophilized 80% ethanol extract of *P. pubescens* leaves were 1.71–11.63 mg/g (Table 1).

## 4. Discussion

In this research, we investigated the effect of PPE on human SRD5A2 promoter activity in human BPH-1 cells and on testosterone-induced BPH in rats. We found that PPE inhibits human SRD5A2 promoter activity and attenuates testosterone-induced prostatic hypertrophy and pathological changes in prostate tissue by reducing the expression of SRD5A2 as well as serum DHT levels, and the results were comparable to those in the finasteride-treated group used as a positive control. Moreover, testosterone-induced expression of AR, PCNA, and TNFα in the prostate was inhibited by PPE treatment. Therefore, these results suggest that PPE has an effective anti-proliferative effect as a therapeutic agent for prostatic hyperplasia.

BPH occurs due to the overgrowth of prostate epithelial cells and stromal cells as men age, which is known to be caused by an imbalance between cell proliferation and apoptosis. DHT is known to be the most important male hormone for prostatic hypertrophy, and SRD5A2 regulates the process of DHT formation from testosterone. For this reason, SRD5A2 has been studied as an important target in the development of many prostate hypertrophy inhibitors. Finasteride, a well-known BPH treatment drug so far, also targets SRD5A2 to reduce blood DHT and reduce prostate size [24]. However, taking finasteride for a long period of time has been known to cause unwanted adverse effects and has warranted the development of a new drug. Therefore, many studies have been conducted to obtain clinical evidence for alternative treatment of BPH. Botanical remedies are drawing attention as part of an effort to replace synthetic drugs and minimize side effects.

Steroid hormones such as testosterone and DHT regulate various cellular processes such as cell growth, proliferation, and differentiation [25]. It is also known to play an important role in the development of BPH by AR, and modulating AR signaling could be a major treatment method for BPH [25]. The testosterone-induced BPH rats had increased serum DHT production and 5α-reductase levels and increased expression of AR in prostate tissue, and its activation may imply interaction with endogenous androgens. However, PPE significantly reduced the expression of AR and showed a significant inhibitory effect on the expression of 5α-reductase consistent with the target-based cellular study that showed PPE inhibited SRD5A2 promoter activity. PCNA is a nuclear protein induced during the G1/S phase transition, acts as a marker for cell proliferation, and plays an important role in physiological conditions such as BPH [26,27]. In this study, the cell proliferation process was compared and analyzed to find out the mechanism of the pathogenesis of BPH using PCNA and FGF2. The Q-PCR and IHF results of PCNA showed that PPE significantly downregulated the expression of PCNA-positive cells in prostate tissue. These observations suggest the anti-proliferative effect of PPE on cell-cycle progressions such as the inhibition of RB/E2F, which is generally associated with the regulation of transcription of DNA synthesis and G1/S regulatory gene activity. Testosterone-induced FGF2 mRNA levels in the prostate were also markedly suppressed by PPE administration, similar to the action of finasteride. Since the regulation of FGF2 is dependent on DHT, inhibition of FGF2 expression by PPE is assumed to be the effect of reduced DHT through inhibition of SRD5A2 expression. These results suggest that PPE may be an effective prostatic hyperplasia treatment.

The area where prostate enlargement occurs is known as the area where chronic inflammation occurs frequently [28]. This chronic inflammatory environment has emerged as one of the important causes of prostatic hyperplasia, and inflammation in prostatic hyperplasia is known to be associated with patient symptoms, the progression of prostate enlargement, and elevated PSA, a prostate indicator protein. Since serum PSA levels are frequently elevated in prostate diseases such as BPH and prostate cancer, they have therefore been used as clinical indicators of disease prognosis [29]. In our study, testosterone-induced BPH rats also showed a significant increase in serum PSA levels. However, oral administration of PPE markedly reduced the serum PSA levels compared to the BPH group, suggesting the prevention of prostatic hypertrophy progression. Since AR occupied by a ligand such as DHT regulates prostate proliferation, while simultaneously upregulating PSA expression, it is thought that the reduction of DHT through PPE-mediated SDR5A2 inhibition consequently suppressed the expression of PSA. A significant reduction in TNFα in the prostate of the PPE treatment group suggests that anti-inflammatory agents are involved in the mechanism of PPE in BPH treatment, but more investigation of other cytokines will be needed to further reinforce our conclusions. Moreover, although further studies of potential chronic toxicity and genotoxicity are required, PPE treatment resulted in no significant increase in serum hepatic parameters, GPT and GOT, which indicates that PPE does not cause hepatotoxicity in rats up to 200 mg/kg of PPE.

*P. pubescens* has been commonly used as a traditional herb in Korea, Japan, and China [30,31,32], and we identified that PPE contained five phytochemicals: chlorogenic acid, isoorientin, orientin, isovitexin, and 4-hydroxycinnamic acid. These ingredients are known to have pharmacological effects including anticancer [33], antioxidant [34], anxiolytic activity [35], and anti-inflammatory [36] effects. Although we have not clearly identified the compounds responsible for the efficacy of PPE, multiple target activities of PPE on BPH suggest that it may involve synergistic or additional effects of several active ingredients in PPE.

## 5. Conclusions

In conclusion, oral administration of PPE prevented and inhibited the development and progression of enlarged prostate lesions in testosterone-induced animal models through various anti-proliferative and anti-inflammatory pharmacological effects as well as through suppression of SRD5A2 gene expression. Although it should be confirmed through human application studies, our findings suggest that PPE may be a potential herbal medicine such as a dietary supplement for BPH therapy that effectively slows the progression of BPH.

## Figures and Tables

**Figure 1 nutrients-13-00884-f001:**
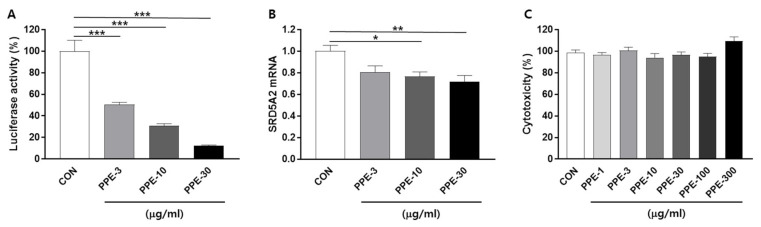
*Phyllostachys pubescens* leaves extract (PPE) suppressed human SRD5A2 promoter activity and mRNA expression. (**A**) Concentration-response effects of PPE on human SRD5A2 promoter activity. BPH-1 cells were transfected with the luciferase reporter plasmid vectors comprising the human SRD5A2 promoter sequence. Cells were treated with the indicated concentrations of PPE for 24 h and luciferase activity was measured by a luminometer. (**B**) mRNA levels of SRD5A2 were measured by Q-PCR. (**C**) Cytotoxicity effects of PPE. All data are shown as mean ± S.E.M. of three independent experiments with triplicate wells. * *p* < 0.05, ** *p* < 0.01, and *** *p* < 0.001, compared with CON.

**Figure 2 nutrients-13-00884-f002:**
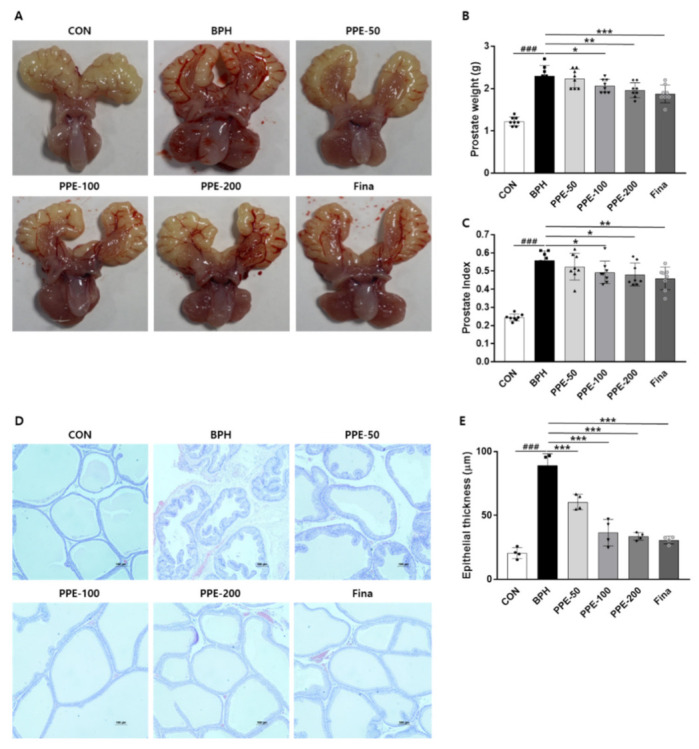
PPE suppressed testosterone-induced prostate enlargement. Castrated rats were given an oral dose of PPE (50, 100, and 200 mg/kg) or finasteride (1 mg/kg) with 5 mg/kg of testosterone injected daily. Rats injected with the vehicle were used as the negative control (CON) group. (**A**) Representative prostate images were shown from each experimental group. (**B**) Prostate weight was measured and analyzed. All data are mean ± SD (*n* = 8 per group). (**C**) The prostatic index was calculated as the ratio of prostate weight to body weight. (**D**) Representative prostatic tissue slide-stained by hematoxylin and eosin (H&E) was shown (magnification × 100). (**E**) The epithelial thickness of prostate was measured and the graph is expressed as the mean ± SD of 4 rats per experimental group.

**Figure 3 nutrients-13-00884-f003:**
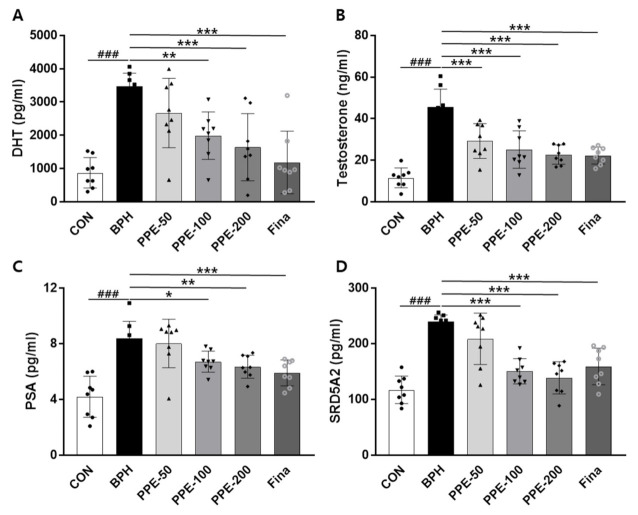
Effect of PPE on DHT, testosterone, PSA, and SRD5A2 levels in serum. The concentration of (**A**) DHT, (**B**) testosterone, (**C**) PSA, and (**D**) SRD5A2 were analyzed using an ELISA kit. Data are presented as mean ± SD (*n* = 8). ### *p* < 0.001 compared with control group, and values with different letters indicate significant differences, * *p* < 0.05, ** *p* < 0.01, and *** *p* < 0.001, compared with BPH group.

**Figure 4 nutrients-13-00884-f004:**
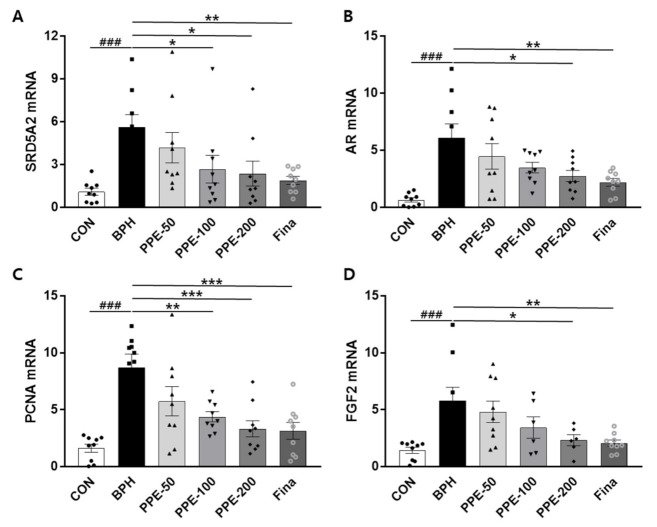
Effect of PPE on mRNA levels of SRD5A2, AR, PCNA, and FGF2 in the prostate. The expression of prostatic genes (**A**) SRD5A2, (**B**) AR, (**C**) PCNA, and (**D**) FGF2 were addressed by real-time quantitative PCR. Data are presented as mean ± SD (*n* = 8). ### *p* < 0.001 compared with control group, and values with different letters indicate significant differences, * *p* < 0.05, ** *p* < 0.01, and *** *p* < 0.001, compared with the BPH group.

**Figure 5 nutrients-13-00884-f005:**
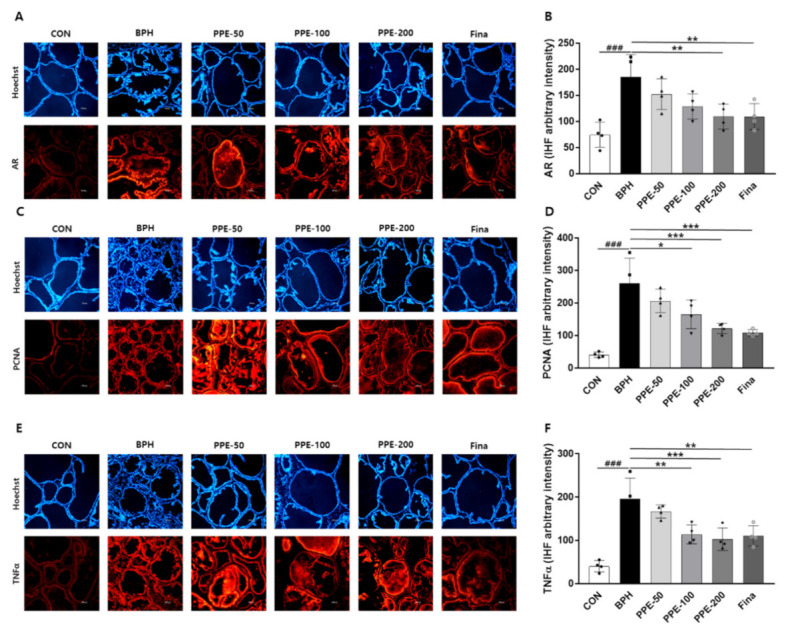
Effect of PPE on AR, PCNA, and TNFα expression in prostate. The expression of (**A**,**B**) AR, (**C**,**D**) PCNA, and (**E**,**F**) TNFα were analyzed by IHF. Representative immunofluorescence images were shown and colabeled with Hoechst 33,258 (blue). Magnification 10×; scale bar, 100 µm. The densities of proteins were calculated using ACT-1 software. Data represent arbitrary intensity relative to CON (control) as the mean ± SD of 4 rats per experimental group. ### *p* < 0.001 compared with control group, and values with different letters indicate significant differences, * *p* < 0.05, ** *p* < 0.01, and *** *p* < 0.001, compared with the BPH group.

**Figure 6 nutrients-13-00884-f006:**
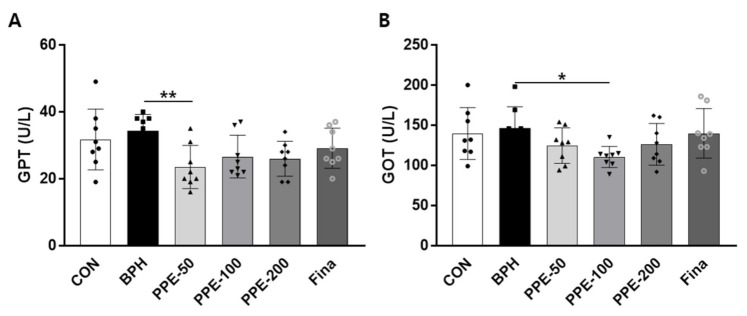
Effect of PPE on serum GTP and GOT levels. (**A**) Serum GPT and (**B**) GOT levels at 4 weeks after PPE treatment are shown. Data are presented as mean ± SD (*n* = 8). Values with different letters indicate significant differences, * *p* < 0.05 and ** *p* < 0.01, compared with the BPH group.

**Figure 7 nutrients-13-00884-f007:**
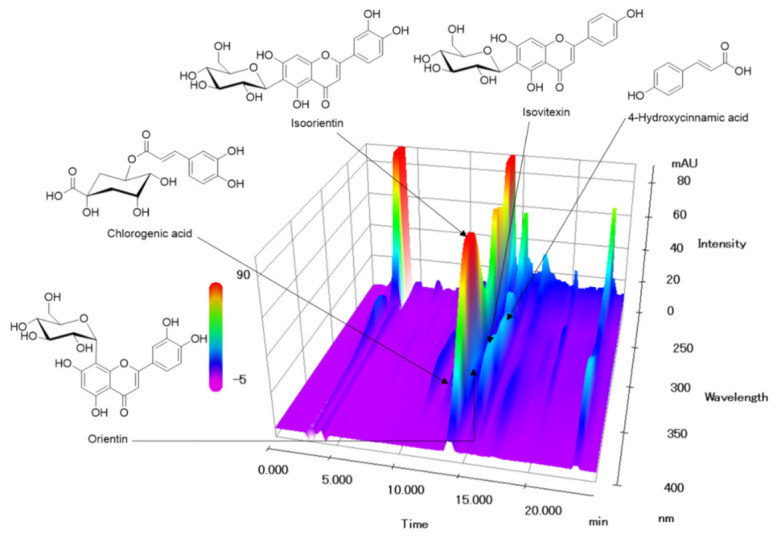
Three-dimensional HPLC chromatogram of *P. pubescens* leaves. Five marker components, chlorogenic acid, isoorientin, orientin, isovitexin, and 4-hydroxycinnamic acid, were detected at 12.24, 13.92, 14.34, 15.30, and 15.66 min, respectively.

**Table 1 nutrients-13-00884-t001:** Amounts of the five marker analytes in *P. pubescens* leaves samples (*n* = 3).

Analyte	Batch no.					
	1		2		3	
	Mean (mg/g) ± SD ^a^ (× 10^−1^)	RSD ^b^ (%)	Mean (mg/g) ± SD (× 10^−1^)	RSD (%)	Mean (mg/g) ± SD (× 10^−1^)	RSD (%)
Chlorogenic acid	1.71 ± 0.08	0.44	1.85 ± 0.16	0.84	1.78 ± 0.17	0.95
Isoorientin	10.94 ± 0.18	0.16	11.63 ± 0.23	0.20	11.29 ± 0.30	0.27
Orientin	3.06 ± 0.03	0.09	3.25 ± 0.09	0.29	3.15 ± 0.02	0.07
Isovitexin	4.36 ± 0.15	0.34	4.65 ± 0.06	0.12	4.51 ± 0.09	0.20
4-Hydroxycinnamic acid	2.13 ± 0.04	0.17	2.27 ± 0.07	0.30	2.20 ± 0.10	0.44

^a^ SD means standard deviation. ^b^ RSD means relative standard deviation and presented as SD/Mean × 100.

## Data Availability

The data presented in this study are available from the corresponding author on reasonable request.

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
