# Peer review of "Extracts of Phyllostachys pubescens Leaves Represses Human Steroid 5-Alpha Reductase Type 2 Promoter Activity in BHP-1 Cells and Ameliorates Testosterone-Induced Benign Prostatic Hyperplasia in Rat Model"

_nutrients, 2021, doi:10.3390/nu13030884_

Round 1

Reviewer 1 Report

In this manuscript authors describe the effect of bamboo leaf extract (PPE) on benign prostate hyperplasia. They observe that PPE represses the activity of 5-alpha-reductase promoter activity and its mRNA expression. PPE treatment also reduces the mRNA levels of AR, PCNA and FGF and the expression of AR, PCNA, and TNF alpha. In animal model, PPE treatment was able to reduce the serum levels of DHT, testosterone, PSA and SRDA5A2. In sum, the oral administration of PPE is responsible for the reduction of enlarged prostate lesions in testosterone treated animal models through antiproliferative and anti-inflammatory effects.

The presented data clearly demonstrate the conclusions but sometimes lack different explanations. For example, authors assert that the expression of AR, PCNA, TNF, FGF and PSA are reduced by PPE treatment, but they don’t explain if this is a direct or indirect effect of PPE. In particular, PSA transcription is regulated by AR but if AR levels are reduced, probably PSA levels are reduced; however, PPE treatment, by reducing 5-alpha reductase levels, decrease the circulating levels of DHT and this is responsible for PSA reduction levels. The authors should comment this for all the others altered proteins.

Author Response

In this manuscript authors describe the effect of bamboo leaf extract (PPE) on benign prostate hyperplasia. They observe that PPE represses the activity of 5-alpha-reductase promoter activity and its mRNA expression. PPE treatment also reduces the mRNA levels of AR, PCNA and FGF and the expression of AR, PCNA, and TNF alpha. In animal model, PPE treatment was able to reduce the serum levels of DHT, testosterone, PSA and SRDA5A2. In sum, the oral administration of PPE is responsible for the reduction of enlarged prostate lesions in testosterone treated animal models through antiproliferative and anti-inflammatory effects.  

Point 1: The presented data clearly demonstrate the conclusions but sometimes lack different explanations. For example, authors assert that the expression of AR, PCNA, TNF, FGF and PSA are reduced by PPE treatment, but they don’t explain if this is a direct or indirect effect of PPE. In particular, PSA transcription is regulated by AR but if AR levels are reduced, probably PSA levels are reduced; however, PPE treatment, by reducing 5-alpha reductase levels, decrease the circulating levels of DHT and this is responsible for PSA reduction levels. The authors should comment this for all the others altered proteins. 

Response 1: We agree with you and have described this suggestion into the discussion section of the manuscript as followed.

Line 382, “These observations suggest the anti-proliferative effect of PPE on cell cycle progressions such as the inhibition of RB/E2F which is generally associated with the regulation of transcription of DNA synthesis and G1/S regulatory genes activity.”

Line 386, “Since the regulation of FGF2 is dependent on DHT, inhibition of FGF2 expression by PPE is assumed to be the effect of reduced DHT through inhibition of SRD5A2 expression.”

Line 399, “In our study, testosterone-induced BPH rats also showed a significant increase in se-rum PSA levels. However, oral administration of PPE markedly reduced the serum PSA levels compared to the BPH group, suggesting the prevention of prostatic hyper-trophy progression. Since AR occupied by a ligand such as DHT regulates prostate proliferation, while simultaneously upregulating PSA expression, it is thought that the reduction of DHT through PPE-mediated SDR5A2 inhibition consequently suppressed the expression of PSA.”

Line 402, “A significant reduction of TNF-α in the prostate of the PPE treatment group suggests that anti-inflammatory agents are involved in the mechanism of PPE in BPH treatment, but more investigation of other cytokines will be needed to further reinforce our conclusions.”

Reviewer 2 Report

The work is interesting. It addresses a problem faced by a large group of men. Sustained results have a great potential for application in the treatment of prostatic hypertrophy

- At what stage were the bamboo leaves used to produce the extracts (young or old)?

- how the leaves were dried (temperature, drying time)

- line 11 is „anesthetized mice”  probably should be „anesthetized rats” 

- Whether the distribution of the results obtained was consistent with a normal distribution?

 - What test was used to check whether the distribution of the results was compatible with normal distribution? Please complete the methodology

line 262 and line 273 is „mice” probably should be „rats”

Author Response

The work is interesting. It addresses a problem faced by a large group of men. Sustained results have a great potential for application in the treatment of prostatic hypertrophy

Point 1: At what stage were the bamboo leaves used to produce the extracts (young or old)?

Response 1: We used naturally air-dried leaves of P. pubescens that are more than 3 years old.

Point 2: how the leaves were dried (temperature, drying time)

Response 2: We used naturally dried leaves of P. pubescens that are more than 3 years old. The drying time varies depending on the condition of the leaves as it is naturally dried instead of hot air drying.

Point 3: line 11 is “anesthetized mice” probably should be “anesthetized rats”

Response 3: We apologize. As reviewer indicated, there was a mistake. We corrected from “mice” to "rats” in line 141.

Point 4: Whether the distribution of the results obtained was consistent with a normal distribution?

Response 4: Thank you for this comment. We evaluated the normality of the distributions using Shapiro–Wilk’s test, and the samples were followed a normal distribution.

Point 5: What test was used to check whether the distribution of the results was compatible with normal distribution? Please complete the methodology

Response 5: As the reviewer suggested, we improved our description of the statistical analysis part of Materials and method section line 202 as follows “Normality of the distributions was evaluated using Shapiro–Wilk’s test. As the samples followed normal distribution,”

Point 6: line 262 and line 273 is “mice” probably should be “rats”

Response 6: As the reviewer indicated, there were typo mistakes. We corrected from “mice” to “rats”. We also correct other typos in lines 198, 208, and 210. We deleted the unnecessary word “Subsection” in line 330.
